# RhoA/Rho-Kinase Signaling in Vascular Smooth Muscle and Endothelium: Mechanistic Insights and Translational Implications in Hypertension

**DOI:** 10.3390/biom15111607

**Published:** 2025-11-16

**Authors:** Stephanie Randar, Diana L. Silva-Velasco, Fernanda Priviero, R. Clinton Webb

**Affiliations:** 1Cardiovascular Translational Research Center, School of Medicine, University of South Carolina, 6311 Garners Ferry Road, Building 1, Columbia, SC 29209, USA; diana.silvavelasco@uscmed.sc.edu (D.L.S.-V.); fernanda.priviero@uscmed.sc.edu (F.P.); 2Department of Biomedical Engineering Molinaroli, Swearingen Engineering Center, College of Engineering and Computing, University of South Carolina, 301 Main Street, Room 3A47, Columbia, SC 29208, USA; 3Department of Cell Biology and Anatomy, School of Medicine, University of South Carolina, 6311 Garners Ferry Road, Building 1, Columbia, SC 29209, USA

**Keywords:** RhoA, Rho kinase, ROS, vascular smooth muscle, endothelium, hypertension, calcium (Ca^2+^) sensitization, remodeling

## Abstract

The small GTPase RhoA and its downstream effector Rho-kinase (ROCK) have emerged as pivotal regulators of vascular smooth muscle cell (VSMC) contraction, endothelial function, and vascular remodeling. Activation of the RhoA/ROCK pathway enhances calcium (Ca^2+^) sensitivity by inhibiting myosin light chain phosphatase (MLCP), thereby promoting sustained vascular tone independent of intracellular Ca^2+^ levels. In endothelial cells (ECs), RhoA/ROCK signaling contributes to nitric oxide (NO) dysregulation, oxidative stress, cytoskeletal reorganization, and inflammatory activation. Cumulative evidence implicates this pathway in the development and progression of hypertension and other cardiovascular diseases, where maladaptive vascular remodeling, VSMC proliferation, and endothelial dysfunction drive increased vascular resistance. Translational studies have identified ROCK inhibitors and indirect modulators such as statins as promising therapeutic strategies. This review integrates recent mechanistic insights into RhoA/ROCK regulation of vascular function with clinical and translational perspectives on targeting this pathway in hypertension.

## 1. Introduction

Hypertension is a leading risk factor for cardiovascular morbidity and mortality, characterized by increased vascular resistance driven by complex interactions among vascular smooth muscle cells (VSMCs), endothelial cells (ECs), and extracellular matrix remodeling [1,2]. Among the multiple signaling networks implicated, the RhoA/Rho-kinase (ROCK) pathway has garnered particular attention for its dual role in regulating vascular tone and driving vascular pathology [3,4,5].

Originally identified as a central regulator of sensitization in smooth muscle contraction, RhoA/ROCK signaling has since been implicated in diverse cellular functions, including cytoskeletal dynamics, proliferation, migration, oxidative stress regulation, and inflammatory signaling [6,7]. Therefore, these mechanisms contribute not only to acute vasoconstriction but also to chronic vascular remodeling, a hallmark of hypertensive pathology.

This review synthesizes mechanistic and translational insights into RhoA/ROCK signaling in VSMCs and ECs, emphasizing its contribution to hypertension and cardiovascular disease, and evaluates current and emerging therapeutic strategies targeting this pathway.

## 2. Mechanistic Pathways of RhoA/ROCK Signaling

### 2.1. Signal Initiation and Ca^2+^ Sensitization

RhoA (Ras homology family member) is a small GTPase highly expressed in VSMCs and is crucial for controlling a range of cellular processes, including contraction [8], proliferation, migration [9], dedifferentiation [10] and phenotypic switching [11]. Most of RhoA’s actions are carried out through its key interactions, particularly with Rho-associated protein kinase (ROCK). The ROCK family consists of two isoforms, ROCK1 (also known as ROKβ or p160ROCK) and ROCK2 (also referred to as ROKα), which share 65% overall sequence identity and 92% identity within the kinase domain. Both kinases feature an N-terminal catalytic kinase domain, followed by a central coiled-coil region that includes a Rho-binding domain (RBD), and a C-terminal pleckstrin-homology domain containing an internal cysteine-rich region [12,13,14]. In both humans and mice, ROCK1 and ROCK2 are expressed ubiquitously across tissues [14]. ROCK sustains contraction primarily by inhibiting myosin light chain phosphatase (MLCP) through phosphorylation of its myosin phosphatase target subunit 1 (MYPT1) and CPI-17, a PKC-potentiated phosphatase inhibitor, thereby sustaining myosin light chain (MLC) phosphorylation and force generation even at constant Ca^2+^ concentrations [15,16,17,18]. In this sense, upon agonist binding to G protein–coupled receptors (e.g., Angiotensin II [AngII], endothelin-1 [ET-1], thrombin), phospholipase C (PLC) activation leads to hydrolysis of phosphatidylinositol 4,5-bisphosphate (PIP_2_) into inositol 1,4,5-trisphosphate (IP_3_) and diacylglycerol (DAG). IP_3_ mobilizes Ca^2+^ from the sarcoplasmic reticulum, while DAG activates protein kinase C (PKC).

RhoA activation is controlled by guanine nucleotide exchange factors (GEFs) that catalyze GDP→GTP exchange; GTPase activating proteins (GAPs) that accelerate GTP hydrolysis; and guanine nucleotide dissociation inhibitors (GDIs), which sequester Rho proteins in the cytosol. Agonists (Ang II, endothelin 1, thrombin, GPCR ligands), mechanical cues (pressure, stretch, shear), and integrin/matrix signals recruit specific GEFs (e.g., p115RhoGEF, LARG) to activate RhoA locally, thereby generating spatially confined contractile responses [12,13]. Notably, intracellular Ca^2+^ elevations are coupled to RhoA activation via PYK2-mediated phosphorylation of PDZ-RhoGEF, which enables Ca^2+^ signals (for example, those triggered by ionophore or receptor activation) to drive downstream RhoA/ROCK responses in VSMCs. Indeed, knockdown of PDZ-RhoGEF or PYK2 markedly suppresses RhoA activation elicited by elevated [Ca^2+^]ᵢ induced by the ionophore A23187 [19]. Recent studies further highlight the role of Pyk2, a Ca^2+^-sensitive non-receptor tyrosine kinase, in coupling membrane depolarization to RhoA activation through regulation of RhoGEFs and RhoGAPs. In rat caudal arterial smooth muscle, Pyk2-mediated RhoA activation facilitates depolarization-induced contraction, emphasizing the importance of upstream Rho regulators in integrating Ca^2+^ signals with RhoA/ROCK activation [20].

Moreover, pharmacological ROCK inhibition attenuates agonist-induced cytosolic Ca^2+^ responses in conductance arteries, such as the aorta and large mesenteric vessels. Interestingly, this effect is less pronounced in small resistance arteries, suggesting vessel-specific modulation of Ca^2+^ signaling by ROCK [21]. Beyond voltage-dependent Ca^2+^ influx, RhoA/ROCK signaling also facilitates non–voltage-gated Ca^2+^ entry pathways, including store-operated and receptor-operated Ca^2+^ influx. This effect may result from ROCK-dependent cytoskeletal reorganization, which enhances coupling between second messenger systems and Ca^2+^ channels [22]. Collectively, these findings support a bidirectional mechanism in which Ca^2+^ signals trigger RhoA/ROCK activation through GEF phosphorylation, while RhoA/ROCK activity, in turn, enhances both Ca^2+^ sensitivity and, in certain contexts, Ca^2+^ entry. This reciprocal regulation likely contributes to the sustained vasoconstriction and vascular tone observed in hypertensive states.

Recent optogenetic studies reinforce this concept by demonstrating that light-activated RhoA can directly induce intracellular Ca^2+^ transients. In HeLa and RPE1 cells, RhoA triggers Ca^2+^ signaling through ROCK–myosin II–dependent activation of mechanosensitive channels, whereas in other epithelial systems such as MDCK cells, it can also signal via the RhoA–PLCε–IP_3_ pathway leading to ER Ca^2+^ release. Such transients activate NFAT nuclear translocation, linking RhoA activity to transcriptional programs and highlighting its role as an integrator of mechanical, Ca^2+^, and gene-regulatory signals [23].

### 2.2. Myosin Light Chain Phosphorylation and VSMC Contraction

Phosphorylation of the regulatory MLC by myosin light chain kinase (MLCK) activates myosin ATPase, enabling actin–myosin cross-bridge cycling. The level of MLC phosphorylation reflects the balance between MLCK and MLCP activities. In this context, RhoA/ROCK enhances MLC phosphorylation primarily through MLCP inhibition and, under certain conditions, via direct phosphorylation of MLC [24]. Furthermore, monophosphorylation at Ser19 and diphosphorylation at Ser19/Thr18 augment actin-activated ATPase activity, strengthening vascular tone [25]. In proliferative or synthetic VSMCs, MLC diphosphorylation is increased, linking phenotypic modulation to vascular remodeling [26]. Collectively, these mechanisms highlight the dual role of RhoA/ROCK in both acute contraction and long-term structural adaptation (Figure 1).

### 2.3. Endothelial Mechanisms in RhoA/ROCK Signaling

In endothelial cells, ROCK activation reduces endothelial nitric oxide synthase (eNOS) expression and activity while increasing endothelial contractility through actomyosin tension. Consequently, cell–cell junctions are disrupted and permeability rises [27]. These changes facilitate leukocyte infiltration and trigger pro-inflammatory signaling, establishing a mechanistic link between endothelial dysfunction and vascular remodeling (Figure 2).

In addition, ROCK modulates endothelial-to-mesenchymal transition (EndMT) and interacts with transcriptional coactivators, such as YAP (Yes, associated protein) and TAZ (Transcriptional coactivator with PDZ-binding motif), contributing to fibrosis and arterial stiffening. These mechanisms contribute to reduced vasodilatory capacity and heightened vascular inflammation in hypertension. Notably, AngII preferentially activates the ROCK2 isoform in endothelial cells, leading to loss of endothelium-dependent relaxation. Isoform-selective inhibition restores endothelial function in both male and female mice, highlighting the therapeutic potential of targeting ROCK2 [28].

## 3. Crosstalk with Oxidative Stress and Redox Signaling

### 3.1. Reactive Oxygen Species (ROS)

ROS such as superoxide anion (O_2_•−) and hydrogen peroxide (H_2_O_2_) are implicated as key regulators of VSMCs function. ROS are produced as byproducts of normal cell metabolism and other enzymatic reactions, particularly via nicotinamide adenine dinucleotide phosphate (NADPH) oxidases (NOX), xanthine oxidase, and the mitochondrial electron transport chain. Major intracellular sources include mitochondria, endoplasmic reticulum, and peroxisomes [29,30]. In addition, exogenous factors such as smoking, pollution, xenobiotics, and radiation further enhance ROS production [31]. At physiological levels, ROS regulate proliferation, differentiation, and apoptosis. However, elevated ROS induces oxidative stress, causing lipid peroxidation, protein oxidation, DNA damage and activation of inflammatory pathways [32]. ROS-induced inflammation results in pathological developments including neurodegenerative, autoimmune, respiratory, and vascular disorders [33]. In the vasculature, elevated ROS levels disrupt cellular homeostasis and promote chronic inflammation, contributing to the pathogenesis of vascular diseases. In VSMCs, oxidative stress impacts Ca^2+^ handling and enhances contractility, which leads to the development of hypertension and vascular remodeling.

### 3.2. Impact on Ca^2+^ Handling and Ca^2+^ Channels

ROS directly and indirectly modulate Ca^2+^ handling in VSMCs by affecting Ca^2+^ channels, pumps, and intracellular Ca^2+^ stores. Given that intracellular Ca^2+^ is the key second messenger mediating VSMCs contractility and vascular tone, these effects have profound physiological consequences [4,5]. For example, ROS regulate a variety of Ca^2+^ channels in the plasma membrane, including voltage-dependent, receptor-induced, and store-mediated. ROS modifies the redox-sensitive cysteine residues on the channel α1-subunit of L-type voltage-gated Ca^2+^ channels, which causes changes in channel trafficking, and ultimately, alterations in VSMC function [34]. The L-type voltage-gated channels in VSMCs, Cav 1.2 channels, are specifically stimulated by ROS produced by NADPH oxidase localized near the plasma membrane. The effect of ROS on these channels was mediated through protein kinase C alpha (PKCα) signaling. H_2_O_2_ has been shown to increase the Ca^2+^ influx through L-type voltage-gated channels in VSMCs. The enhanced stimulation of Cav 1.2 channels and increased influx of Ca^2+^ causes sustained vasoconstriction [34]. Oxidative stress and ROS can impact Ca^2+^ handling in VSMCs by influencing the expression and function of ion channels, such as transient receptor potential melastatin 2 (TRPM2) which is a ROS sensor and regulator of Ca^2+^ and Na^+^ transport. During hypertensive states, ROS production can be enhanced, which increases the activation of TRPM2 channels on VSMCs via the Poly [ADP-ribose] polymerase 1 (PARP1) protein [35]. This leads to increased Ca^2+^ influx through a Ca^2+^/Na^+^ exchanger, leading to increased intracellular Ca^2+^ and VSMC contraction. An example of store operated Ca^2+^ entry is the Orai1 channel on plasma membranes in VSMCs. In physiological conditions, the Orai pathway increases Ca^2+^ influx when Ca^2+^ stores in the endoplasmic reticulum are depleted. The sensor protein, STIM1, activates Orai1 channels, allowing more Ca^2+^ to enter the cell. In hypertensive conditions, high ROS levels enhance the Orai1 channel function and there is excessive Ca^2+^ entry into the cell. This leads to vascular dysfunction and sustained vasoconstriction [36].

### 3.3. Ca^2+^ Microdomains

Ca^2+^ microdomains in VSMCs are small, localized regions of elevated Ca^2+^ concentrations formed near specific signaling receptors and ion channels on the plasma membrane or endoplasmic reticulum [37]. These microdomains are organized via caveolin scaffolding proteins within caveolae, which cluster receptors and channels in defined areas. In this sense, the localized high Ca^2+^ concentrations enable precise activation of downstream signaling pathways without significantly altering global intracellular Ca^2+^, ensuring tightly controlled vascular contraction and relaxation [38]. Importantly, Ca^2+^ microdomains play a crucial role in balancing vascular tone and are often involved in pathological conditions. In hypertensive conditions, Ang II activates NADPH oxidase, leading to excess ROS in the body. Consequently, ROS alter the Ca^2+^ microdomains by over activating certain receptors and reorganizing the receptor expression in the caveolae. For example, Cav1.2 channels can be upregulated within microdomains, enhancing Ca^2+^ influx and contributing to sustained vasoconstriction, which maintains hypertension [39]. Moreover, hyperglycemia-induced ROS in VSMCs disrupt caveolar signaling, which not only affects Ca^2+^ microdomains organization but also leads to dysregulated expression of adipokines such as adiponectin and leptin. Although adiponectin is primarily secreted by adipocytes, its synthesis in VSMCs appears to be a compensatory response mediated by ROS and NADPH oxidase type 4 (Nox4) within caveolae, while downregulation of adiponectin receptors may limit protective effects. These caveola-dependent changes in both Ca^2+^ signaling and adipokine expression further exacerbate vascular dysfunction under metabolic stress [40].

Beyond Ca^2+^ handling, localized Ca^2+^ signals within caveolae can activate RhoA via Ca^2+^/calmodulin-dependent pathways and GEFs, which subsequently stimulate ROCK to enhance Ca^2+^ sensitization of contractile responses. Therefore, dysregulation of Ca^2+^ microdomains under oxidative stress may potentiate RhoA/ROCK activation, further contributing to vascular hypercontractility in hypertension [41].

### 3.4. Redox Sensitive Proteins

Redox sensitive proteins serve as a critical bridge between ROS and Ca^2+^ signaling in VSMCs, influencing vascular tone, contractility, and cellular function. Several key proteins, including PARP1, protein tyrosine phosphatases (PTP), protein kinases, and transcription factors, are involved in these redox-regulated pathways. Specifically, PARP1 is activated by oxidative DNA damage in VSMCs and catalyzes the synthesis of ADP-ribose polymers, which are broken down into free ADP ribose (ADPR). ADPR directly binds to the C-terminal Nudix-type motif 9 homologous (NUDT9-H) domain of TRPM2 channels, inducing a conformational change that opens the channel and facilitates Ca^2+^ and Na^+^ influx [42]. Consequently, VSMC contractility increases, particularly in hypertensive states where ROS levels are elevated. In physiological conditions, PTPs act to balance the protein tyrosine kinases (PTKs) by dephosphorylating tyrosine residues on proteins [43]. However, excessive ROS oxidizes cysteine residues within PTPs, leading to their inactivation. Therefore, unchecked phosphorylation of signaling proteins leads to increased Ca^2+^ influx and heightened vascular tone [44]. Two examples of redox sensitive protein kinases include Ca^2+^/calmodulin-dependent protein kinase II (CaMKII) and phosphatidylinositol 3-kinase (PI3K) proteins. CaMKII is directly oxidized by ROS through the methionine residue, leading to sustained kinase activity without sustained stimulation. This prolonged kinase activity causes vascular stiffening and VSMC apoptosis [45]. PI3K, which normally regulates VSMC phenotype, including contraction and differentiation [46], shows increased membrane recruitment and catalytic activity upon H_2_O_2_ stimulation. This enhances ROS accumulation and activates the AKT signaling pathway, ultimately altering VSMCs. The combined results lead to alterations in VSMC phenotype [47]. In addition, under oxidative stress, specific methionine residues (Met281/282) within CaMKII’s regulatory domain undergo oxidation, leading to sustained kinase activation independent of Ca^2+^/calmodulin binding [48]. This persistent activation promotes VSMC apoptosis, hypertrophic signaling, and vascular stiffening. In cardiac tissue, oxidized CaMKII has been implicated in arrhythmogenesis, myocardial injury, and maladaptive remodeling, linking redox imbalance to Ca^2+^-driven vascular and cardiac pathology [49]. ROS also stimulates a variety of transcription factors with protective effects over VSMCs to combat the damage done through the previously mentioned pathways. The redox-sensitive transcription factor, nuclear factor erythroid 2-related factor 2 (Nrf2) is activated by oxidative stress and functions to increase the transcription of antioxidants, which protect against oxidative stress [50]. In contrast, chronic high levels of ROS shifts the balance toward harmful transcription factors, such as nuclear factor-kappa B (NF-kB), which promotes secretion of pro-inflammatory cytokines, leads to further production of ROS, and ultimately causes vascular damage [51]. Collectively, redox-sensitive proteins integrate oxidative signals with Ca^2+^ dynamics and downstream transcriptional programs, establishing a feedback loop that amplifies vascular dysfunction under hypertensive conditions.

Collectively, these redox-sensitive proteins not only amplify oxidative stress and Ca^2+^ disturbances but also converge on RhoA/ROCK activation. For instance, oxidation-driven activation of PTPs enhances RhoA activity through sustained phosphorylation of RhoA-associated kinases, while oxidized CAMKII can indirectly promote cytoskeletal remodeling via ROCK signaling [7,52]. This integration of redox and RhoA/ROCK pathways represents a central mechanism linking oxidative stress to vascular stiffness and remodeling.

## 4. RhoA/ROCK in Hypertension and Vascular Diseases

### 4.1. Increased Contractility and Peripheral Vascular Resistance

In hypertension, increased contractility of VSMC leads to sustained peripheral vascular resistance and vascular dysfunction [53] (Figure 3). This heightened contractility results from both elevated intracellular Ca^2+^ levels and Ca^2+^ sensitization mechanisms. In particular, activation of the RhoA/ROCK pathway decreases MLCP activity, thereby sustaining vasoconstriction and elevating blood pressure even in the presence of lower Ca^2+^ concentrations. Additionally, an imbalance in redox signaling, often characterized by excessive ROS production from NOX enzymes, contributes to increased VSMC contractility and elevated peripheral vascular resistance. Elevated ROS levels during hypertensive states activate multiple signaling molecules through oxidation of specific protein residues. Consequently, kinases such as PKC and ROCK are hyperactivated resulting in heightened Ca^2+^ sensitivity and sustained vasoconstriction of blood vessels. In addition, ROS alter the function of Ca^2+^ channels and pumps, such as the L-type Ca^2+^ channels, enhancing influx of Ca^2+^ into VSMCs. Moreover, RhoA/ROCK signaling amplifies endothelium- and endoperoxide-dependent contractile responses, which are hallmarks of hypertensive vascular dysfunction [54]. Overall, ROS increases VSMC contractile capabilities and leads to prolonged hypertension by enhancing peripheral vascular resistance.

Consistent with these mechanisms, animal studies have demonstrated elevated ROCK activity primarily in VSMCs across multiple hypertensive models, including spontaneously hypertensive rats (SHRs), Ang II–infused, deoxycorticosterone acetate (DOCA)-salt, and Dahl salt-sensitive rats [55]. Pharmacological inhibition of ROCK in VSMCs with agents such as Y-27632 or fasudil significantly reduces vascular tone, prevents medial thickening, and attenuates the rise in blood pressure [56,57]. These findings highlight the mechanistic role of RhoA/ROCK signaling in maintaining VSMCs contractility and structural remodeling in hypertension.

Supporting this, rat models of apatinib-induced hypertension exhibit upregulation of RhoA and ROCK2 in the mid-aorta, accompanied by reduced MLCP activity and increased ET-1 and collagen I expression. Administration of the ROCK inhibitor Y-27632 significantly reduced both blood pressure and vascular remodeling, confirming the critical role of ROCK2 in contractile dysfunction [58]. Furthermore, genetic studies provide further evidence for the role of RhoA/ROCK signaling in vascular contractility. In Dahl salt-sensitive rats, loss of the Rho GEF *Arhgef11* attenuates ROCK activation, decreased vascular contractility, and protection against hypertension-induced renal injury. This highlights *Arhgef11* as an upstream regulator of the RhoA/ROCK pathway in hypertension [59].

### 4.2. RhoA/ROCK-Induced Endothelial Dysfunction in Hypertension

Endothelial dysfunction is a characteristic of vascular disease and is often mediated by oxidative stress and Ca^2+^ dysregulation. Vascular endothelium, a monolayer of cells lining the lumen, regulates both permeability and vascular tone. In this context, ROS impair NO production by oxidizing and inactivating eNOS, thereby reducing vasodilation and increasing vascular tone [60] (Figure 3). Additionally, abnormal Ca^2+^ release from intracellular stores and enhanced entry into endothelial cells promote pathological reactions. Increased cytosolic Ca^2+^ levels can over activate enzymes promoting oxidative stress and inflammatory signals. In addition, high Ca^2+^ levels contribute to vascular calcification, which can harden VSMCs and further contribute to vascular dysfunction, atherosclerosis and morbidity. Calcification involves stiffening of vessel walls which limits the vasodilation capabilities, increasing the peripheral vascular resistance [60].

### 4.3. RhoA/ROCK-Mediated Suppression of NO Bioavailability and Vascular Remodeling

RhoA/ROCK signaling contributes to endothelial dysfunction and vascular remodeling through multiple, interconnected mechanisms. ROCK downregulates eNOS transcription, destabilizes eNOS mRNA, decreases Ser1177 phosphorylation, and enhances oxidative stress, leading to eNOS uncoupling [61,62]. In addition, increased arginase activity further reduces L-arginine availability, collectively these effects impair endothelium-dependent relaxation. Supporting this, exposure of endothelial cells to high-salt conditions activates RhoA/ROCK signaling and suppresses eNOS expression through the upregulation of asymmetric dimethylarginine (ADMA), an endogenous eNOS inhibitor. This highlights dietary salt as an upstream modulator of the RhoA/ROCK–NO axis in vascular dysfunction [63].

Moreover, ROCK-mediated EC contraction which is referring to actomyosin-driven-cytoskeletal tension that modulates cell–cell junctions [64], increases permeability and promotes leukocyte adhesion/transmigration. This is reinforced by activation of NF-κB–dependent adhesion molecules (VCAM-1, ICAM-1) and chemokines, linking endothelial ROCK activity to vascular inflammation [65].

In parallel, vascular remodeling occurs through alterations in vessel thickness, lumen diameter, and elasticity within the wall [66]. In VSMCs, ROS-driven phenotypic modulation, growth, death, migration, and extracellular matrix reorganization underlie these structural changes. Excessive ROS contributes to vascular wall thickening, lumen narrowing, and reduced elasticity. In addition, matrix malloproteinases (MMPs) involved in extracellular matrix remodeling are influenced by ROS, leading to disorganized remodeling, increased collagen deposition, and vessel stiffening [67]. The phenotypic changes caused by ROS are associated with altered cell growth and apoptosis, leading to hyperplasia and hypertrophy of VSMCs. At lower levels, ROS induces VSMC growth. However, at high levels, ROS dampen protective properties of VSMC, causing unnecessary apoptosis, contributing to atherosclerosis and atypical vascular remodeling [68]. Importantly, ROS in VSMCs not only drive these structural changes but also directly modulate RhoA/ROCK signaling. Oxidative stress activates RhoA through redox-sensitive GEFs such as p115RhoGEF [69], enhancing ROCK-mediated phosphorylation of MLC and promoting VSMCs contraction and stiffness [70]. Thus, ROS-induced remodeling is closely linked to RhoA/ROCK-dependent cytoskeletal reorganization.

Notably, studies in normotensive rats with elevated angiotensin-converting enzyme levels revealed increased ROCK activation in the aortic wall, alongside upregulation of profibrotic and inflammatory genes such as monocyte chemoattractant protein-1, transforming growth factor (TGF)-beta(1) (TGF-β1), plasminogen activator inhibitor-1 (PAI-1), and monocyte chemoattractant protein-1 (MCP-1). This was accompanied by enhanced NADPH oxidase activity and superoxide production, all of which were normalized by the ROCK inhibitor fasudil, highlighting the role of RhoA/ROCK in early vascular remodeling even before the onset of hypertension [71].

Finally, by augmenting Ca^2+^ sensitivity, RhoA/ROCK sustains vasoconstriction independently of changes in intracellular Ca^2+^ concentration, a phenomenon observed in hypertensive resistance vessels [72]. In contrast, RhoA/ROCK signaling in the cardiomyocytes promotes hypertrophic gene expression via activation of serum response factor (SRF) and extracellular signal-regulated kinase (ERK). The RhoA/ROCK signaling in the cardiomyocytes leads to maladaptive outcomes such as biventricular hypertrophy and myocardial stiffening as a result of hypertensive stress [73,74].

## 5. Inflammation and Immune Activation

ROS play a pivotal role in promoting vascular inflammation in endothelial cells and VSMCs and activating immune cells, thereby influencing vascular function and accelerating tissue damage. Inflammation not only results from vascular dysfunction but also perpetuates it, creating a self-sustaining pathological loop. Elevated ROS levels activate redox-sensitive transcription factors such as NF-kB which induces the expression of pro-inflammatory cytokines, chemokines, and adhesion molecules [75]. This, in turn, promotes the recruitment and activation of immune cells at the endothelial level, facilitating their adhesion and subsequent transmigration into the surrounding connective tissue, where they amplify local oxidative stress and inflammation. The infiltration of immune cells, particularly macrophages and neutrophils, further amplifies oxidative stress by generating additional ROS through respiratory bursts, aggravating endothelial injury [76]. Moreover, elevated intracellular Ca^2+^ concentrations can activate inflammasomes, such as NOD-, LRR- and pyrin domain-containing protein 3 (inflammasome) (NLRP3), in immune cells leading to the release of pro-inflammatory cytokines that exacerbate vascular inflammation [77]. Activation of NLRP3 in VSMCs promotes a phenotypic switch from contractile to synthetic and proliferative states via NF-κB signaling and increased IL-1β and IL-18 secretion, driving vascular hypertrophy, stiffness, and elevated peripheral resistance. In endothelial cells, NLRP3 activation reduces eNOS activity and NO bioavailability, further promoting endothelial dysfunction [78,79]. Experimental evidence supports these effects, as NLRP3 activation in VSMCs and ECs has been shown to induce vascular inflammation remodeling, and impaired vasodilation in cardiovascular diseases [80,81,82]. Moreover, under oxidative stress, NLRP3- mediated inflammation enhances the canonical RhoA/ROCK signaling pathway [83], amplifying Ca^2+^ sensitization, cytoskeletal reorganization, and vascular contraction. In addition, continuous ROS production in VSMCs and endothelial cells maintains a chronic inflammatory state, establishing a vicious cycle that contributes to progressive vascular damage, characteristic of hypertension and atherosclerosis [84].

## 6. Human Evidence

Evidence from human studies corroborates the preclinical findings. In mononuclear cells from patients with essential hypertension, p63RhoGEF mRNA and protein levels, as well as phospho-myosin phosphatase target protein-1 (MYPT-1), a marker of Rho kinase activity, were significantly increased compared with healthy individuals, indicating enhanced RhoA/ROCK signaling in vivo [85]. Acute fasudil infusion improves forearm blood flow and reduces arterial stiffness in humans [86] supporting the concept that ROCK inhibition can restore endothelial function. However, long-term clinical trials evaluating chronic blood pressure-lowering effects remain limited.

Further clinical evidence shows that cigarette smokers display significantly increased leukocyte ROCK activity, which correlates with higher systolic blood pressure and impaired flow-mediated dilation (FMD) [87]. This association suggests that activated leukocytes may contribute to endothelial dysfunction through increased adhesion to the vascular wall, potentially promoting oxidative stress and inflammatory signaling, and subsequent endothelial injury. Thus, leukocyte ROCK activation may reflect systemic oxidative stress and is associated with endothelial dysfunction in smokers [87]. Consistent with these findings, young adults with metabolic syndrome exhibit increased RhoA/ROCK activity along with elevated markers of endothelial dysfunction, even before the onset of overt hypertension [72]. This clinical evidence highlights the early involvement of RhoA/ROCK in vascular pathology and its potential as a therapeutic target in at-risk populations.

## 7. Therapeutic Targeting

### 7.1. No Pan-ROCK Inhibitors Are in Clinical Use for the Treatment of Hypertension

Systemic ROCK inhibition can cause severe hypotension, headache, edema, and off-target effects (gastrointestinal, urinary, etc.), reflecting broad ROCK distribution [88]. Additionally, the lack of isoform selectivity (ROCK1 vs. ROCK2) complicates therapeutic precision [89]. As a result, clinical development for ROCK inhibitors has shifted towards targeted and local applications to modulate VSMCs and endothelial cell function. For instance, fasudil is used for cerebral vasospasm following subarachnoid hemorrhage associated with hypertension, while topical formulations of netarsudil (Rhopressa) and ripasudil (Glanatec) are employed for glaucoma and ocular hypertension [90].

### 7.2. Emerging Therapeutic Strategies

Many of the emerging strategies for disrupting RhoA/Rho-kinase signaling in the vasculature are highly experimental and caution is needed to address off-target effects and potential adverse events associated with pan-ROCK inhibition. Promising next-generation approaches for the treatment of hypertension include isoform-selective inhibitors, such as KD025, which exhibits antifibrotic properties in preclinical studies using animal model of hypertension [91], nanoparticle vascular delivery [92], and targeting upstream activators (specific RhoGEFs) [93].

Additionally, recent work has identified the TRPV4–RhoA–RhoGDI1 axis as a key regulatory pathway in vascular tone. Activation of TRPV4 promotes RhoA inactivation through RhoGDI1, and pharmacologic targeting of this axis reduces vascular contractility and lowers blood pressure in hypertensive models. These findings suggest that enhancing endogenous inhibitory mechanisms of RhoA may represent a complementary therapeutic approach [94].

The RhoA/ROCK signaling axis plays a central role in VSMCs, calcium sensitization, arterial constriction and structural remodeling in hypertension [7]. However, ROCK1 and ROCK2 are ubiquitously expressed in multiple tissues, including heart, skeletal muscle, kidney, lung, and immune cells and participate in a broad range of cellular functions such as cytoskeletal organization, cell motility, proliferation/apoptosis, matrix remodeling, and endothelial barrier regulation [95,96]. Thus, the widespread tissue distribution and pleiotropic roles of ROCK1/2 represent a major challenge for the development of vascular-specific therapeutic interventions, as systemic inhibition may produce unwanted effects in non-vascular tissues. To overcome this, recent strategies have focused on isoform-selective inhibition [97] and on tissue/cell-type-specific targeting, with preclinical studies demonstrating favorable effects on vascular remodeling and blood pressure with reduced non-vascular toxicity [95,96]. These developments suggest that while vascular specificity of ROCK targeting remains an obstacle, tailored approaches with isoform specificity or targeted delivery hold promise to enhance therapeutic efficacy and safety in hypertension.

## 8. Future Directions

Future research on RhoA/ROCK inhibitors in hypertension focus on improving therapeutic precision, identifying specific disease mechanisms, and optimizing clinical use [89]. Research is moving beyond pan-inhibition to define isoform-specific roles (ROCK1 vs. ROCK2) within VSMCs and ECs to develop more targeted therapies and avoid systemic side effects. Mapping the upstream Rho GEFs and GAPs activated in hypertensive phenotypes is crucial for understanding disease-specific activation of the pathway [89,98]. Simultaneously, the development of reliable biomarkers, such as phospho-MYPT1 or leukocyte ROCK activity, will allow for better patient stratification and monitoring of treatment response in clinical practice [99,100]. To enhance efficacy and reduce monotherapy risks, combination therapies are being explored, particularly combining ROCK inhibitors with established treatments like RAAS blockade or antioxidants [101]. Ultimately, conducting robust, mechanism-based clinical trials is necessary to assess the long-term safety and efficacy of chronic ROCK inhibition and solidify its role in managing hypertension. In summary:Define isoform-specific contributions (ROCK1 vs. ROCK2) in VSMCs vs. ECs.Map upstream RhoGEFs and GAPs activated in hypertensive phenotypes.Develop biomarkers (phospho-MYPT1, leukocyte ROCK activity) for patient stratification.Explore combination therapies (ROCK inhibition + RAAS blockade or antioxidants).Conduct mechanism-based clinical trials assessing chronic ROCK inhibition in hypertension.

## 9. Conclusions

The RhoA/ROCK pathway represents a central regulator of vascular tone, endothelial function, and remodeling. Mechanistic insights into Ca^2+^ sensitization, MLCP inhibition, oxidative stress amplification, and endothelial dysfunction underscore its role in hypertension pathogenesis. Translational studies highlight ROCK inhibitors and statins as promising therapeutic approaches, though broader clinical application requires careful balancing of efficacy and safety. Precision targeting RhoA/ROCK signaling offers a compelling strategy for future antihypertensive therapies and vascular protection.

## Figures and Tables

**Figure 1 biomolecules-15-01607-f001:**
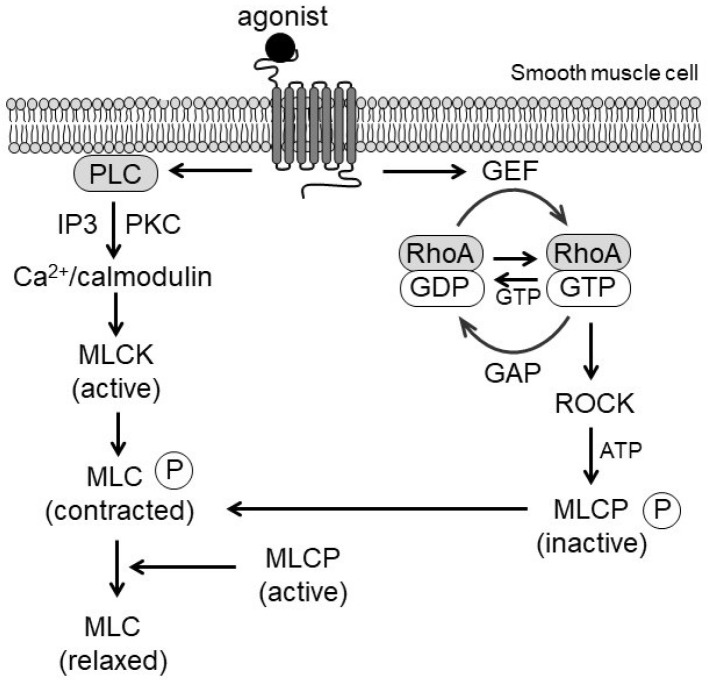
Schematic representation of Ca^2+^/calmodulin and RhoA/ROCK signaling in vascular smooth muscle contraction. Agonist binding to G protein-coupled receptors (GPCRs) activates phospholipase C (PLC) and guanine nucleotide exchange factors (GEFs), leading to inositol trisphosphate (IP_3_)–mediated Ca^2+^ release, myosin light chain kinase (MLCK) activation, and phosphorylation of myosin light chain (MLC). Concurrently, RhoA activation stimulates Rho-kinase (ROCK), which inhibits myosin phosphatase (MLCP) through phosphorylation of its myosin-binding subunit, enhancing Ca^2+^ sensitization and sustaining vascular contraction.

**Figure 2 biomolecules-15-01607-f002:**
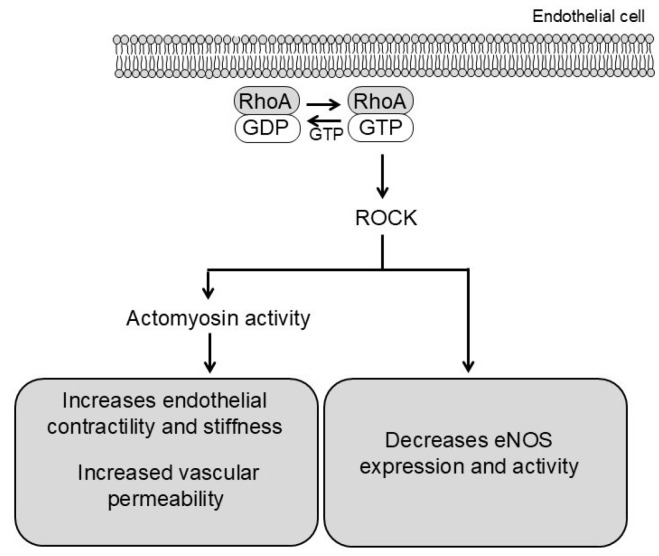
Schematic representation of ROCK signaling in endothelial cells. ROCK exerts two related effects: (1) it inhibits eNOS activity via intracellular signaling pathways, reducing nitric oxide (NO) production and increasing vascular tone; and (2) it directly enhances actomyosin contractility, increasing endothelial contractility (modulation of cell–cell junctions) and permeability.

**Figure 3 biomolecules-15-01607-f003:**
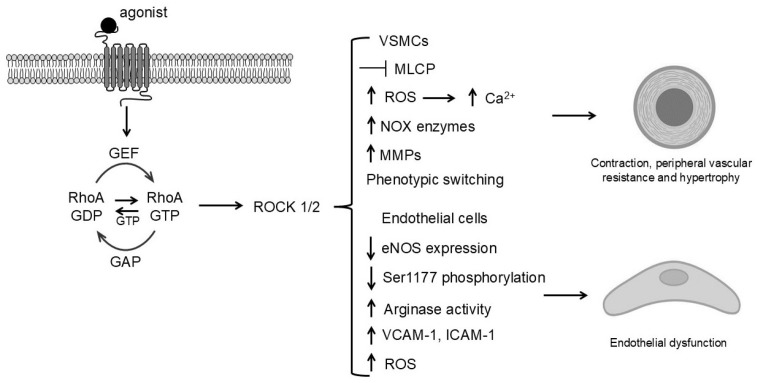
Mechanisms of RhoA/ROCK-mediated vascular cell proliferation and remodeling during hypertension. Agonist binding to G protein-coupled receptors (GPCRs) leading to ROCK1/2 activation in VSMCs and endothelial cells. In VSMCs, ROCK enhances MLC phosphorylation, cytoskeletal remodeling, and phenotypic switching, resulting in proliferation, migration, and extracellular matrix deposition. In endothelial cells, ROCK decreases eNOS expression and increases permeability and inflammation. Collectively, these mechanisms drive vascular wall thickening, stiffness, and reduced lumen diameter, contributing to elevated peripheral resistance and hypertension.

## Data Availability

No new data were created or analyzed in this study.

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
