# Peer review of "RhoA/Rho-Kinase Signaling in Vascular Smooth Muscle and Endothelium: Mechanistic Insights and Translational Implications in Hypertension"

_biomolecules, 2025, doi:10.3390/biom15111607_

Round 1
Reviewer 1 Report
Comments and Suggestions for Authors
This review provides a comprehensive and well-written analysis of RhoA/ROCK regulation in vascular smooth muscle and endothelial cells, linking mechanistic insights to therapeutic strategies for alleviating hypertension-associated vascular disease.
However, certain modifications are necessary to enhance clarity in specific sections.
Paragraph 2.1.
Lines 100-106. For clarity, the authors should specify the particular cell lines or cellular systems in which the described activities of RhoA were experimentally demonstrated.
Paragraph 2.2.
Figure 1 could be clearer if modified as in the attached Figure 1. Also, in both the figure and the caption, myosin phosphatase should be indicated as it is abbreviated in the text (MLCP).
Paragraph 2.3.
What is stated in paragraph 2.3 is correct, but it is not properly explained in Figure 2. ROCK exerts two related effects: 1) it suppresses nitric oxide synthesis by reducing eNOS activity (which normally promotes vascular relaxation through the production of nitric oxide in endothelial cells, thereby inducing relaxation of the underlying vascular smooth muscle), thus contributing to increased vascular tone; and 2) it increases actomyosin contractile tension in endothelial cells, thereby enhancing both endothelial contractility and permeability. The inhibitory effect of ROCK on eNOS is mediated by intracellular signaling pathways and not directly by actomyosin contraction itself. Therefore, Figure 2 should be modified to take this explanation and the corrections shown in the accompanying figure into account.
Paragraph 3.3.
Lines 204-207. The demonstration that hyperglycemia induces an increase in adiponectin synthesis in VSMCs should be explained better, as this is very interesting, especially considering that this hormone is produced mainly by adipocytes.
Paragraph 4.
I think it would be more correct to write 'vascular diseases' instead of 'cardiovascular diseases' in the title of paragraph 4, since the paragraph does not mention cardiac diseases.
Paragraph 4.1.
Line 271. Replace ‘guanine nucleotide exchange factor‘ with GEF.
Paragraph 5.
Lines 330-331. I would specify better, for example: ‘promotes the recruitment and activation of immune cells at the endothelial level, leading to their infiltration into the surrounding connective tissue.’
Paragraph 6.1.
It should be explained in which cells elevated ROCK activity has been demonstrated.
Paragraph 6.2.
How does increased ROCK activity in leukocytes of cigarette smokers correlate with endothelial dysfunction? Is it because leukocyte infiltration toward the vascular endothelium is increased, and these cells, by producing pro-inflammatory cytokines and ROS, directly damage endothelial cells? This should be explained.

Author Response
This review provides a comprehensive and well-written analysis of RhoA/ROCK regulation in vascular smooth muscle and endothelial cells, linking mechanistic insights to therapeutic strategies for alleviating hypertension-associated vascular disease. However, certain modifications are necessary to enhance clarity in specific sections.
Comment 1: Paragraph 2.1. Lines 100-106. For clarity, the authors should specify the particular cell lines or cellular systems in which the described activities of RhoA were experimentally demonstrated.
Response 1: We thank the reviewer for this suggestion. We have now specified the cellular systems in which the RhoA-mediated Ca²⁺ signaling was observed. Specifically, the optogenetic studies were performed in HeLa and RPE1 cells, where RhoA triggers Ca²⁺ transients via ROCK–myosin II–dependent activation of mechanosensitive channels. In addition, in MDCK cells, RhoA was shown to signal through the PLCε–IP₃ pathway leading to ER Ca²⁺ release. We have updated the manuscript accordingly to improve clarity (see paragraph 2.1. lines 100-106) “Recent optogenetic studies reinforce this concept by demonstrating that light-activated RhoA can directly induce intracellular Ca²⁺ transients. In HeLa and RPE1 cells, RhoA triggers Ca²⁺ signaling through ROCK–myosin II–dependent activation of mechanosensitive channels, whereas in other epithelial systems such as MDCK cells, it can also signal via the RhoA–PLCε–IP₃ pathway leading to ER Ca²⁺ release. Such transients activate NFAT nuclear translocation, linking RhoA activity to transcriptional programs and highlighting its role as an integrator of mechanical, Ca²⁺, and gene-regulatory signals (Inaba, H., Q. Miao, and T. Nakata, 2021, J Biol Chem. 296: p. 100290)”.
Comment 2: Paragraph 2.2. Figure 1 could be clearer if modified as in the attached Figure 1. Also, in both the figure and the caption, myosin phosphatase should be indicated as it is abbreviated in the text (MLCP).
Response 2: We sincerely appreciate this comment and the time taken to modify the figure. We have implemented these changes in both the figure and the text.
Comment 3: Paragraph 2.3. What is stated in paragraph 2.3 is correct, but it is not properly explained in Figure 2. ROCK exerts two related effects: 1) it suppresses nitric oxide synthesis by reducing eNOS activity (which normally promotes vascular relaxation through the production of nitric oxide in endothelial cells, thereby inducing relaxation of the underlying vascular smooth muscle), thus contributing to increased vascular tone; and 2) it increases actomyosin contractile tension in endothelial cells, thereby enhancing both endothelial contractility and permeability. The inhibitory effect of ROCK on eNOS is mediated by intracellular signaling pathways and not directly by actomyosin contraction itself. Therefore, Figure 2 should be modified to take this explanation and the corrections shown in the accompanying figure into account.
Response 3: We thank the reviewer for the insightful comment. We agree that distinguishing the two related effects of ROCK in the figure improves clarity. Using the figure, you proposed we believe that these modifications address the concern and better illustrate the mechanistic link described in the text. In the revised version, you will find the newly proposed figure, as well as its corresponding figure legend.
Comment 4: Paragraph 3.3. Lines 204-207. The demonstration that hyperglycemia induces an increase in adiponectin synthesis in VSMCs should be explained better, as this is very interesting, especially considering that this hormone is produced mainly by adipocytes.
Response 4: We thank the reviewer for this insightful comment. We have clarified the text to better explain the hyperglycemia-induced increase in adiponectin synthesis in VSMCs. We have included this information in the new version as follows: paragraph 3.3. lines 206-214: “Moreover, hyperglycemia-induced ROS in VSMCs disrupt caveolar signaling, which not only affects Ca²⁺ microdomains organization but also leads to dysregulated expression of adipokines such as adiponectin and leptin. Although adiponectin is primarily secreted by adipocytes, its synthesis in VSMCs appears to be a compensatory response mediated by ROS and NADPH oxidase type 4 (Nox4) within caveolae, while downregulation of adiponectin receptors may limit protective effects. These caveola-dependent changes in both Ca²⁺ signaling and adipokine expression further exacerbate vascular dysfunction under metabolic stress (El Atab, O., et al., 2022 Eur J Pharmacol 919: p. 174701)”.
Comment 5: Paragraph 4. I think it would be more correct to write 'vascular diseases' instead of 'cardiovascular diseases' in the title of paragraph 4, since the paragraph does not mention cardiac diseases.
Response 5: Thank you for your suggestion. We have made the corresponding change.
Comment 6: Paragraph 4.1. Line 271. Replace ‘guanine nucleotide exchange factor‘ with GEF.
Response 6: Thank you for your suggestion. We have made the corresponding change. It appears in line 297.
Comment 7: Paragraph 5. Lines 330-331. I would specify better, for example: ‘promotes the recruitment and activation of immune cells at the endothelial level, leading to their infiltration into the surrounding connective tissue.’
Response 7: We thank the reviewer for this suggestion. We have clarified the text to specify that ROS-induced endothelial activation promotes adhesion and recruitment of immune cells at the endothelial surface, followed by their transmigration into the surrounding connective tissue. This more accurately reflects the mechanism described in the literature, where infiltrating immune cells amplify oxidative stress and inflammation within the vascular wall. The revised sentence (lines 375-378) now reads: “This, in turn, promotes the recruitment and activation of immune cells at the endothelial level, facilitating their adhesion and subsequent transmigration into the surrounding connective tissue, where they amplify local oxidative stress and inflammation.”
Comment 8: Paragraph 6.1. It should be explained in which cells elevated ROCK activity has been demonstrated.
Response 8: We appreciate the reviewer’s comment. We have now specified that elevated ROCK activity has been demonstrated primarily in VSMCs in hypertensive models, including SHR, Ang II–infused, DOCA-salt, and Dahl salt-sensitive rats. The revised text clarifies the cellular context in which ROCK contributes to increased vascular tone and remodeling. We have included this information in the new version as follows (lines 283-290): “Consistent with these mechanisms, animal studies have demonstrated elevated ROCK activity primarily in VSMCs across multiple hypertensive models, including spontaneously hypertensive rats (SHR), Ang II–infused, deoxycorticosterone acetate (DOCA)-salt, and Dahl salt-sensitive rats (Wirth, A, 2010, Biochim Biophys Acta. 1802(12): p. 1276-84). Pharmacological inhibition of ROCK in VSMCs with agents such as Y-27632 or fasudil significantly reduces vascular tone, prevents medial thickening, and attenuates the rise in blood pressure (Seko, T., et al. 2003. Circ Res. 92(4): p. 411-8. Uehata, M., et al., 1997. Nature, 389(6654): p. 990-4). These findings highlight the mechanistic role of RhoA/ROCK signaling in maintaining VSMCs contractility and structural remodeling in hypertension”.
Comment 9: Paragraph 6.2. How does increased ROCK activity in leukocytes of cigarette smokers correlate with endothelial dysfunction? Is it because leukocyte infiltration toward the vascular endothelium is increased, and these cells, by producing pro-inflammatory cytokines and ROS, directly damage endothelial cells? This should be explained.
Response 9: We thank the reviewer for this insightful comment. As discussed by Hidaka et al. (2010), increased ROCK activity in peripheral leukocytes of smokers correlates inversely with endothelial function (assessed by FMD). While the study did not directly measure ROS or inflammatory cytokines, it suggests that leukocyte ROCK activation may enhance adhesion and activation at the endothelial surface, contributing to endothelial dysfunction. Additionally, leukocyte ROCK activity likely reflects ROCK signaling in vascular endothelial cells, which can impair NO bioavailability and vasodilatory function. We have revised the text to clarify this mechanism as follows (lines 406-412): “Further clinical evidence shows that cigarette smokers display significantly increased leukocyte ROCK activity, which correlates with higher systolic blood pressure and impaired flow-mediated dilation (FMD) (Hidaka, T., et al., 2010. Hypertens Res. 33(4): p. 354-9). This association suggests that activated leukocytes may contribute to endothelial dysfunction through increased adhesion to the vascular wall, potentially promoting oxidative stress and inflammatory signaling, and subsequent endothelial injury. Thus, leukocyte ROCK activation may reflect systemic oxidative stress and is associated with endothelial dysfunction in smokers (Hidaka, T., et al., 2010. Hypertens Res. 33(4): p. 354-9).
Reviewer 2 Report
Comments and Suggestions for Authors
In this review article (ID# biomolecules-3968325), entitled “RhoA/Rho-Kinase Signaling in Vascular Smooth Muscle and Endothelium: Mechanistic Insights and Translational Implications in Hypertension”, the authors Randar et al. summarize the role of vascular RhoA/ROCK signaling in the regulation of vascular function and the development of hypertension, and discuss its potential clinical applications as a therapeutic target. This topic is important since hypertension remains a major risk factor for cardiovascular diseases. The article is generally well written. However, I have several recommendations for improvement listed below.
- RhoA/ROCK plays an important role in regulating vascular function and in the pathogenesis of hypertension. However, as intracellular signaling molecules, they are also expressed in various other tissues and have broad biological functions. Therefore, a major challenge for their clinical use as drug targets lies in their lack of vascular specificity. This issue should be discussed in more detail.
- On page 7, line 298, please clarify the term “EC contraction.” Similarly, in Figure 2, the term “endothelial contractility” needs further explanation. How do endothelial cells contract? Please elaborate on the underlying mechanism or provide supporting references.
- In lines 302–312, the role of ROS in vascular remodeling is discussed in an entire paragraph. Likewise, the sections “3.3. Ca2+ microdomains” and “3.4. Redox-sensitive proteins” include substantial discussion of topics not directly related to RhoA/ROCK signaling. The authors are encouraged to focus more specifically on RhoA/ROCK-related mechanisms throughout the article.
- It would strengthen the review to include a diagram illustrating the mechanisms of RhoA/ROCK-mediated vascular cell proliferation and vascular remodeling during hypertension, similar as contractility regulation in figure 1.
- Please specify the cell type (vascular smooth muscle cells vs. endothelial cells) when discussing each mechanism or process, since these cell types play distinct roles in vascular regulation and the development of hypertension.
- The article’s structure could be improved for better readability. For example, the section on animal models could be integrated into other relevant sections, as most of the review’s content is derived from animal studies. It may not be necessary to have a separate section solely for animal models.
Author Response
In this review article (ID# biomolecules-3968325), entitled “RhoA/Rho-Kinase Signaling in Vascular Smooth Muscle and Endothelium: Mechanistic Insights and Translational Implications in Hypertension”, the authors Randar et al. summarize the role of vascular RhoA/ROCK signaling in the regulation of vascular function and the development of hypertension, and discuss its potential clinical applications as a therapeutic target. This topic is important since hypertension remains a major risk factor for cardiovascular diseases. The article is generally well written. However, I have several recommendations for improvement listed below.
Comment 1: RhoA/ROCK plays an important role in regulating vascular function and in the pathogenesis of hypertension. However, as intracellular signaling molecules, they are also expressed in various other tissues and have broad biological functions. Therefore, a major challenge for their clinical use as drug targets lies in their lack of vascular specificity. This issue should be discussed in more detail.
Response 1: We thank the reviewer for highlighting the important point regarding the major challenge for vascular-specific therapeutic targeting. In response, we have expanded the discussion in the revised manuscript to emphasize that ROCK1 and ROCK2 are ubiquitously expressed in multiple tissues and participate in diverse cellular processes, making systemic inhibition prone to off-target effects. We also highlight current strategies to overcome these limitations, including isoform-selective inhibition and tissue- or cell-type-specific targeting, underscoring that tailored approaches hold promise to enhance both efficacy and safety in hypertension (see lines 442–455 of the revised manuscript): “The RhoA/ROCK signaling axis plays a central role in VSMCs, calcium sensitization, arterial constriction and structural remodeling in hypertension (Seccia, T.M., et al., 2020. J Clin Med. 9(5)). However, ROCK1 and ROCK2 are ubiquitously expressed in multiple tissues, including heart, skeletal muscle, kidney, lung, and immune cells and participate in a broad range of cellular functions such as cytoskeletal organization, cell motility, proliferation/apoptosis, matrix remodeling, and endothelial barrier regulation (Dong, M., et al., 2010. Drug Discov 15(15-16): p. 622-9; Hartmann, S., A.J. Ridley, and S. Lutz., 2015. Front Pharmacol,6: p. 276). Thus, the widespread tissue distribution and pleiotropic roles of ROCK1/2 represent a major challenge for the development of vascular-specific therapeutic interventions, as systemic inhibition may produce unwanted effects in non-vascular tissues. To overcome this, recent strategies have focused on iso-form‑selective inhibition (Fayed, H.S., et al., 2023. Int J Mol Sci, 24(19)) and on tissue/cell‑type‑specific targeting, with pre‑clinical studies demonstrating favorable effects on vascular remodeling and blood pressure with reduced non‑vascular toxicity (Dong, M., et al., 2010. Drug Discov 15(15-16): p. 622-9; Hartmann, S., A.J. Ridley, and S. Lutz., 2015. Front Pharmacol,6: p. 276). These developments suggest that while vascular specificity of ROCK targeting remains an obstacle, tailored approaches with isoform specificity or targeted delivery hold promise to enhance therapeutic efficacy and safety in hypertension.”
Comment 2: On page 7, line 298, please clarify the term “EC contraction.” Similarly, in Figure 2, the term “endothelial contractility” needs further explanation. How do endothelial cells contract? Please elaborate on the underlying mechanism or provide supporting references.
Response 2: Thank you for this insightful comment. In the revised manuscript we have clarified the term “EC contraction” and the related phrase “endothelial contractility” in Figure 2, by explicitly describing the mechanisms by which endothelial cells generate contractile forces and how this contributes to increased permeability and leukocyte adhesion/transmigration. More specifically, we have added a short explanation that endothelial contraction refers to actomyosin‑mediated cytoskeletal tension within the endothelial monolayer, driven by activation of the RhoA/ROCK signaling axis, which promotes stress‑fiber formation (via RhoA‑ROCK‑MLC phosphorylation), increased cell–cell junctional tension and hence enhanced permeability. We cite work showing that thrombin‑induced endothelial permeability requires Rho/ROCK‑mediated actomyosin contractility downstream of Gα12/13 signaling. With this addition, Figure 2 legend and associated text now clearly define “endothelial contractility”. We have included this information in the new version as follows (lines 324-328): “Moreover, ROCK-mediated EC contraction which is referring to actomyosin-driven-cytoskeletal tension that modulates cell-cell junctions (Gavard, J. and J.S. Gutkind, 2008. J Biol Chem. 283(44): p. 29888-96), increases permeability and promotes leukocyte adhesion/transmigration. This is reinforced by activation of NF-κB–dependent adhesion molecules (VCAM-1, ICAM-1) and chemokines, linking endothelial ROCK activity to vascular inflammation (Gimbrone, M.A., Jr. and G. Garcia-Cardena,2016. Circ Res. 118(4): p. 620-36).”
Comment 3: In lines 302–312, the role of ROS in vascular remodeling is discussed in an entire paragraph. Likewise, the sections “3.3. Ca2+ microdomains” and “3.4. Redox-sensitive proteins” include substantial discussion of topics not directly related to RhoA/ROCK signaling. The authors are encouraged to focus more specifically on RhoA/ROCK-related mechanisms throughout the article.
Response 3: We appreciate the reviewer’s valuable comment. We have revised the sections concerning ROS, Ca²⁺ microdomains, and redox-sensitive proteins to emphasize their mechanistic relevance to RhoA/ROCK signaling. Specifically, we now discuss how ROS activate RhoA through redox-sensitive GEFs, how Ca²⁺ microdomains can modulate RhoA/ROCK-dependent Ca²⁺ sensitization, and how redox-sensitive proteins such as CaMKII and PTPs converge on RhoA/ROCK-mediated cytoskeletal remodeling. This information is found in the new manuscript on the lines 339-344 “Importantly, ROS in VSMCs not only drive these structural changes but also directly modulate RhoA/ROCK signaling. Oxidative stress activates RhoA through redox-sensitive GEFs such as p115RhoGEF (Chen, Z., et al., 2012. J Biol Chem. 287(30): p. 25490-500) enhancing ROCK-mediated phosphorylation of MLC and promoting VSMCs contraction and stiffness (Eckenstaler, R., M. Hauke, and R.A. Benndorf. 2022. Biochem Pharmacol, 206: p. 115321). Thus, ROS-induced remodeling is closely linked to RhoA/ROCK-dependent cytoskeletal reorganization.”; lines 215-219 “Beyond Ca²⁺ handling, localized Ca²⁺ signals within caveolae can activate RhoA via Ca²⁺/calmodulin-dependent pathways and GEFs, which subsequently stimulate ROCK to enhance Ca²⁺ sensitization of contractile responses. Therefore, dysregulation of Ca²⁺ microdomains under oxidative stress may potentiate RhoA/ROCK activation, further contributing to vascular hypercontractility in hypertension (Webb, R.C.,2003. Adv Physiol Educ. 27(1-4): p. 201-6).”; lines 259-264 “Collectively, these redox-sensitive proteins not only amplify oxidative stress and Ca²⁺ disturbances but also converge on RhoA/ROCK activation. For instance, oxidation-driven activation of PTPs enhances RhoA activity through sustained phosphorylation of RhoA-associated kinases, while oxidized CAMKII can indirectly promote cytoskeletal re-modeling via ROCK signaling (Seccia, T.M., et al., 2020. J Clin Med. 9(5); Meng, T.C., T. Fukada, and N.K. Tonks. 2002. Mol Cell. 9(2): p. 387-99). This integration of redox and RhoA/ROCK pathways represents a central mechanism linking oxidative stress to vascular stiffness and remodeling.”
Comment 4: It would strengthen the review to include a diagram illustrating the mechanisms of RhoA/ROCK-mediated vascular cell proliferation and vascular remodeling during hypertension, similar as contractility regulation in figure 1.
Response 4: We appreciate this valuable suggestion. In response, we added a schematic diagram (Figure 3) summarizing the mechanisms of RhoA/ROCK-mediated vascular cell proliferation and remodeling during hypertension. This figure illustrates the upstream stimuli, intracellular signaling events, and downstream structural and functional changes in vascular smooth muscle and endothelial cells that contribute to vascular dysfunction and hypertension.
Comment 5: Please specify the cell type (vascular smooth muscle cells vs. endothelial cells) when discussing each mechanism or process, since these cell types play distinct roles in vascular regulation and the development of hypertension.
Response 5: We have revised the sections to specify the cell type (VSMC or EC) associated with each mechanism. These clarifications were added throughout the text (highlighted) to emphasize the complementary but distinct roles of both vascular cell types in RhoA/ROCK-mediated signaling during hypertension. The specific changes can be found in the new manuscript at lines 101, 286, 330, 340, 346, 370–371, 382, 402-404, and 424. We have also carefully reviewed the mechanisms proposed in the hypertension section, and the suggested figure has been updated to visually distinguish these mechanisms between VSMCs and ECs.
Comment 6: The article’s structure could be improved for better readability. For example, the section on animal models could be integrated into other relevant sections, as most of the review’s content is derived from animal studies. It may not be necessary to have a separate section solely for animal models.
Response 6: We thank the reviewer for this valuable comment, which has helped improve the flow and readability of our manuscript. In response, we have removed the separate section on animal models and have integrated the relevant preclinical data into the sections where it best supports the discussion of vascular mechanisms and translational relevance. These integrations can be found in the revised manuscript in lines 283-290: “Consistent with these mechanisms, animal studies have demonstrated elevated ROCK activity primarily in VSMCs across multiple hypertensive models, including spontaneously hypertensive rats (SHR), Ang II–infused, deoxycorticosterone acetate (DOCA)-salt, and Dahl salt-sensitive rats (Wirth, A., 2010. Biochim Biophys Acta. 1802(12): p. 1276-84). Pharmacological inhibition of ROCK in VSMCs with agents such as Y-27632 or fasudil significantly reduces vascular tone, prevents medial thickening, and attenuates the rise in blood pressure (Seko, T., et al., 2003. Circ Res. 92(4): p. 411-8; Uehata, M., et al., 1997. Nature. 389(6654): p. 990-4). These findings highlight the mechanistic role of RhoA/ROCK signaling in maintaining VSMCs contractility and structural remodeling in hypertension.”
Reviewer 3 Report
Comments and Suggestions for Authors
Review article discuss clinically relevant and actual issue dealing with the implication of GTPase RhoA/Rho/kinase signalling in vascular smooth muscle and endothelium regulation in hypertension. The latter is dominant risk factor for cardiovascular diseases that deteriorate to heart failure contributing to high morbidity and mortality. Authors aimed to pay attention to the recent mechanistic insight to RhoA/ROCK regulation of vascular function as possible targeting pathway in hypertension. It should be emphasized that RhoA/ROCK signalling has been implicated in various cellular function including regulation of oxidative stress and inflammation. While these are crucial factors implicated in worsening cardiovascular function and disease progression. Noteworthy, authors pay attention to current therapeutic strategy targeting RhoA/ROCK regulation. Undoubtedly, the article is challenging to pay attention GTPase RhoA/Rho/kinase signalling and its regulation in cardiovascular protection.
The article is clearly and precisely written providing relevant comprehensive information supported by related references and illustrated figures. Paper may be interesting for readers and motivate investigation.
Questions:
Knowing that meta inflammation is general feature of cardiovascular diseases, including hypertension, it may be expected that NLRP3 inflammasome is involved in modulation of GTPase RhoA/Rho/kinase signalling in vascular smooth muscle and endothelium regulation.
Are there some data about implication of NLRP3 inflammasome in modulation of vascular smooth muscle and endothelium in cardiovascular diseases? As well as involvement GTPase RhoA/Rho/kinase signalling?
Is it possible to differentiate GTPase RhoA/Rho/kinase signalling in vascular smooth muscle and endothelium from heart muscle in hypertension?
Author Response
Review article discuss clinically relevant and actual issue dealing with the implication of GTPase RhoA/Rho/kinase signaling in vascular smooth muscle and endothelium regulation in hypertension. The latter is dominant risk factor for cardiovascular diseases that deteriorate to heart failure contributing to high morbidity and mortality. Authors aimed to pay attention to the recent mechanistic insight to RhoA/ROCK regulation of vascular function as possible targeting pathway in hypertension. It should be emphasized that RhoA/ROCK signaling has been implicated in various cellular function including regulation of oxidative stress and inflammation. While these are crucial factors implicated in worsening cardiovascular function and disease progression. Noteworthy, authors pay attention to current therapeutic strategy targeting RhoA/ROCK regulation. Undoubtedly, the article is challenging to pay attention GTPase RhoA/Rho/kinase signaling and its regulation in cardiovascular protection. The article is clearly and precisely written providing relevant comprehensive information supported by related references and illustrated figures. Paper may be interesting for readers and motivate investigation.
Comment 1: Knowing that meta inflammation is general feature of cardiovascular diseases, including hypertension, it may be expected that NLRP3 inflammasome is involved in modulation of GTPase RhoA/Rho/kinase signaling in vascular smooth muscle and endothelium regulation.
Response 1: We thank the reviewer for this feedback. We agree that meta-inflammation is a general feature of cardiovascular diseases, in response, we have now added a clarifying section regarding the NLRP3 inflammasome and its role in cardiovascular disease through interaction with vascular smooth muscle cells and endothelial cells. These additions can be found in the revised manuscript in lines 384-388: “Activation of NLRP3 in VSMCs promotes a phenotypic switch from contractile to synthetic and proliferative states via NF-κB signaling and increased IL-1β and IL-18 secretion, driving vascular hypertrophy, stiffness, and elevated peripheral resistance. In endothelial cells, NLRP3 activation reduces eNOS activity and NO bioavailability, further promoting endothelial dysfunction (Krishnan, S.M., et al., 2019. Cardiovasc Res. 115(4): p. 776-787; Zheng, Y., et al., 2022. Front Cardiovasc Med, 9: p. 927061).”
Comment 2: Are there some data about implication of NLRP3 inflammasome in modulation of vascular smooth muscle and endothelium in cardiovascular diseases? As well as involvement GTPase RhoA/Rho/kinase signaling?
Response 2: We thank the reviewer for their insightful comment and interest in the potential crosstalk between NLRP3 inflammasome activation and RhoA/ROCK signaling in vascular cells. There is indeed experimental evidence supporting the role of NLRP3 in modulating vascular smooth muscle and endothelial cell function in cardiovascular diseases, including its contribution to vascular inflammation, phenotypic switching, endothelial dysfunction, and impaired vasodilation. We have incorporated this evidence into the revised manuscript to clarify the involvement of NLRP3 in both VSMCs and ECs and its interplay with RhoA/ROCK signaling (see revised section 5, lines 388–389): “Experimental evidence supports these effects, as NLRP3 activation in VSMCs and ECs has been shown to induce vascular inflammation remodeling, and impaired vasodilation in cardiovascular diseases (Bai, B., et al., 2020. Cell Death Dis. 11(9): p. 776; Burger, F., et al., 2021. Int J Mol Sci. 23(1); Sun, J., et al., 2023. Atherosclerosis. 387: p. 117391).Moreover, under oxidative stress, NLRP3-mediated inflammation enhances the canonical RhoA/ROCK signaling pathway (Nunes, K.P. and R.C. Webb, 2021. 12(5-6): p. 458-469.), amplifying Ca²⁺ sensitization, cytoskeletal reorganization, and vascular contraction.
Comment 3: Is it possible to differentiate GTPase RhoA/Rho/kinase signaling in vascular smooth muscle and endothelium from heart muscle in hypertension?
Response 3: We thank the reviewer for highlighting this important aspect. Indeed, experimental evidence indicates that RhoA/ROCK signaling exhibits tissue-specific roles in hypertension. In our revised manuscript, we have included this information to clarify that in VSMCs, RhoA/ROCK enhances calcium sensitization and inhibits MLCP, sustaining vasoconstriction, while in ECs it disrupts cytoskeletal integrity and reduces eNOS activity, promoting endothelial dysfunction. In cardiomyocytes, by contrast, RhoA/ROCK primarily drives hypertrophic remodeling via SRF and ERK activation, resulting in myocardial stiffening and biventricular hypertrophy. This distinction is now highlighted in the revised manuscript (lines 355–359): “In contrast, RhoA/ROCK signaling in the cardiomyocytes promotes hypertrophic gene ex-pression via activation of serum response factor (SRF) and extracellular signal-regulated kinase (ERK). The RhoA/ROCK signaling in the cardiomyocytes leads to maladaptive outcomes such as biventricular hypertrophy and myocardial stiffening as a result of hypertensive stress (Dai, Y., W. Luo, and J. Chang. 2018. Curr Opin Physiol, 1: p. 14-20; Okamoto, R., et al., 2013. FASEB J; 27(4): p. 1439-49).”
Round 2
Reviewer 2 Report
Comments and Suggestions for Authors
The revised manuscript has been improved significantly, no further recommendation.